# Understanding Vision and Language Representations under the Lens of Intrinsic Dimension

## Abstract

Current multimodal representation learning is mainly achieved by intuitive and heuristic approaches. However, the cooperation and the utility of each modality remain unclear. We empirically investigate the intrinsic dimension (ID) of a large-scale vision-language pre-training model BLIP and explore the relationships among intrinsic dimension, modality, and prunability. It is shown that the ID geometric characteristics of visual and language representations differ significantly in terms of range and shape, resulting in distinct prunability for each modality. Unified multimodal learning can be manifested as the overlay of ID variations of vision and language. By investigating the IDs of attention representations, it is evident that the current cross-modal attention mechanism struggles to embed modalities into the same low-dimensional manifold due to the varying levels of IDs between vision and language. Moreover, We study the contribution of different modalities toward model prunability and explore predicting model performance through the distributions of IDs. An importance metric based on ID is proposed, which yields superior performance for multimodal model pruning. The experimental results show that visual representations are more sensitive and fragile to pruning, while language representations are robust and, therefore, have a higher prunability. 90% BLIP weights in language modality can be pruned with only 3.8 drops on the CIDEr metric. Our observations suggest the potential for more effective pruning of multimodal models using intrinsic dimension (ID) as a guiding metric.

## 1 Introduction

Multimodal pre-training integrates large-scale vision and language data to learn cross-modal representations. Various multimodal pre-training approaches are proposed to capture the semantic alignment and interaction between vision and language, such as ViLBERT (Lu et al., 2019), LXMERT (Tan & Bansal, 2019), and CLIP (Radford et al., 2021). These models adopt different levels of multimodal alignment and learning strategies. However, all these works rely on an implicit hypothesis that different modalities share a common semantic manifold, where concepts with similar semantics are close or even overlapping. Therefore, cross-modal representation learning can be achieved by measuring the similarity between different modalities. On the other hand, with the increasing number of parameters in pre-training models, these types of modality alignment (Wang et al., 2023; Dong et al., 2021; Liu et al., 2021a) have become more and more sophisticated and fine-grained. Although these methods lead to better generalization performance on downstream tasks, however, the fundamental hypothesizes of the common semantic manifold remain unexplored: Are the representations of vision and language evidently connected in the same semantic space? Specifically, how does the cross-modal attention mechanism, as fundamental for multimodal pre-training, capture information beyond mere vector matching? Moreover, how do the varying parameter counts of different modalities contribute to the overall model performance?

To explore the above questions, it is necessary to have a unified and essential metric to quantify semantics. Although the high-dimensional representations are already a semantic abstraction of training data, their dimensions are manually fixed (e.g.,512, 1024) and thus cannot fully reflect the intrinsic semantic representations. The manifold hypothesis (Bengio et al., 2013) states that high-

dimensional data of interest often live in an unknown lower-dimensional manifold embedded in ambient space, which enables the intrinsic dimension (ID) to be a further abstract representation of semantics. Unlike the feature dimension of networks, which is usually set to 512, 1024, and 2048 in common vision tasks. The estimated ID is generally below 150 for popular visual datasets (Pope et al., 2021; Brown et al., 2023) and visual representations (Ansuini et al., 2019). In this study, ID is adopted as a primary metric to observe the geometrical properties and their correlation with pre-training performance on image captioning tasks.

Current work on intrinsic dimension mainly focuses on the unimodal, particularly the visual modality. Ansuini et al. (2019) investigates the ID profile of three common CNN-based pre-trained representations and finds the hunchback shape of the ID variation across the layers. Muratore et al. (2022) observes similar first-expansion-then-reduction of object representations along the rat homolog of the ventral stream. Pope et al. (2021) estimates the intrinsic dimensionality of several popular datasets and finds that common natural image datasets have very low intrinsic dimensions relative to the high number of pixels in the images. ID is used to study the semantic complexity of synthetic images by GAN (Pope et al., 2021; Horvat & Pfister, 2022; Barannikov et al., 2021), which allows actively manipulating the intrinsic dimension by controlling the image generation process. Brown et al. (2023) empirically verify the hypothesis of the union of manifolds in common image datasets and find that the data lies on a disconnected set with varying intrinsic dimensions. Amsaleg et al. (2017); Ma et al. (2018) use the local ID to characterize the adversarial robustness of attacked visual regions and find that the LID increases along with the increasing noise in adversarial perturbations.

Compared with the large number of ID studies on visual information, including datasets and representations, there are fewer studies on the ID characteristics of language modality. Fine-tuning of the large language model (BERT (Kenton & Toutanova, 2019) and RoBERTa (Liu et al., 2019)) are analyzed from the ID perspective in Aghajanyan et al. (2020). Both theoretical and empirical explanations have been provided, pointing to a low-dimensional reparameterization that is as effective in fine-tuning as the full parameter space. Kvinge et al. (2023) focuses on the prompts for text-to-image generation. It demonstrates that prompt variations affect the intrinsic dimension of model layers in distinct ways. Bottleneck layers, instead of latent layers, correlate with prompt perplexity and intrinsic dimension. Tulchinskii et al. (2023) find that the average intrinsic dimensionality of fluent texts in natural language hovers around the value of 7 to 9 for human-generated texts, while the average intrinsic dimensionality of AI-generated texts for each language is around 1.5 or even lower. The clear statistical separation enables a simple classifier to distinguish human-generated and AI-generated texts.

The aforementioned works are predominantly focused on unimodality. However, our work delves deeper into the large-scale vision-and-language pre-training model through the lens of intrinsic dimension. It provides a comprehensive understanding of cross-modal representations and their distinctive prunability. Specifically, the main contributions of this work are as follows.

- To the best of our knowledge, this work presents the first empirical study into the ID of a large-scale multimodal pre-training model. The geometric properties of IDs for visual and language representations are significantly different. IDs of visual modality are varied with a hunchback shape, ranging from 29 to 180. In contrast, IDs of language modality are uniform with a lower range from 5 to 30. Cross-modal learning can be manifested by the overlay of these two ID variations.

- This work provides a detailed explanation of how visual and language modalities align and change IDs in cross-modal attention mechanisms. We argue that the visual and language representations do not lie on the same low-dimensional manifold. cross-modal attention struggles with embedding low-dimensional language representations into high-dimensional visual manifolds.

- We explore the correlation between IDs and layer-wise importance for multimodal pruning. The experiment results demonstrate that utilizing the ID as an indicator for weight pruning yields a superior compression rate and model performance. Notably, pruning either vision or language modality leads to different changes in IDs. Language representations are more robust with higher prunability, while vision representations are more sensitive and have a greater impact on overall performance.

## 2 ESTIMATING IDs OF THE BLIP PRE-TRAINING

### 2.1 BLIP MODEL

We use BLIP as a surrogate model to investigate the characteristics of multimodal representations. BLIP (Bootstrapping Language-Image Pre-training) (Li et al., 2022) is a framework for vision-language pre-training (VLP) that extends CLIP (Contrastive Language-Image Pre-training) (Radford et al., 2021), a contrastive learning method that learns from noisy web image-text pairs. Unlike CLIP, which only uses an encoder-based model, BLIP introduces a multimodal mixture of encoder-decoder (MED) architecture that can flexibly transfer to both understanding-based and generation-based tasks. It filters out noisy captions with a bootstrapping strategy in a CapFilt module and trains with three objectives: image-text contrastive learning, image-text matching, and image-conditioned language modeling. BLIP utilizes large-scale web data and human-annotated data to provide diverse and effective representations.

BLIP consists of two unimodal streams: a vision model implemented by ViT (Dosovitskiy et al., 2022) and a language model implemented by BERT (Devlin et al., 2018), respectively. The two streams are connected by multimodal attention layers that allow cross-modal alignment by closing their difference in a shared embedding space. In our implementation, we use BLIP finetuned versions with ViT-Base/16 and CapFilt-Large.

### 2.2 TWONN

TwoNN algorithm (Facco et al., 2017) is applied to measure the intrinsic dimension (ID) of the BLIP representations in each layer. ID is a measurement of the effective degrees of freedom or the information content of a set of data that can reveal the complexity and structure of the underlying low-dimensional manifold. The TwoNN algorithm is a simple yet robust and efficient method based on the ratio of distances to the nearest neighbors.

Methods that rely on assumptions about the density or smoothness of the data manifold may not yield accurate estimates of the intrinsic dimension (ID) for high-dimensional data. However, the TwoNN algorithm provides a viable alternative by estimating the ID based solely on local information derived from nearest neighbors. This approach is impervious to scaling and rotation, and does not require any parametric assumptions regarding the distribution of the data. Furthermore, it avoids the need for complex optimization procedures, rendering it more flexible and efficient than the Maximum Likelihood Estimation (MLE) (Levina & Bickel, 2004) method and other approaches that rely on local eigenvalues or geometric properties.

The intrinsic dimensionality is estimated by TwoNN through analyzing the distance between each point and its first and second nearest points. The distance between them is represented by $r_1$ and $r_2$, respectively. The quotient of these two distances, known as $\mu$, is always less than 1 by definition. However, $\mu$ increases as the ID increases. $\mu$ follows a Pareto distribution $Pa(d + 1)$ where $d$ is the intrinsic dimension. The likelihood of multiplying sample of this distribution $\mu = (\mu_1, \mu_2, ...\mu_N)$ can be calculated by

$$P(\mu|d) = d^N \prod_{i=1}^{N} \mu_i^{-d-1}$$

.

Data samples can be obtained by evaluating the network representations on a certain dataset and solving it as a linear regression problem. This method avoids assumptions on the global distribution other than the density is const around each point. This makes it a good fit for the estimation of ID for high-dimensional representations.

### 2.3 APPLY TWONN ON BLIP

Our implementation follows Ansuini et al. (2019) to perform the empirical study ID statistics of BLIP representations in each layer. The BLIP model consists of 12 blocks for both vision and language modalities, corresponding to ViT and BERT model, respectively. With the input of image and caption pairs, the activations of each layer is treated as a data point in its own linear space. ID of each layer is estimated separately under the same dataset MSCOCO (Lin et al., 2014). Since

the time and space complexity of TwoNN is $O(n_D^2)$, the size of dataset $n_D$ is critical for effective estimation. Facco et al. (2017) empirically suggest to use around 10 times the intrinsic dimension. In our implementation, 2,000 samples are used.

The ID estimation of BLIP representations is shown in Figure 1, which shows distinct distributions across vision and language representations. Detailed analysis of ID and multimodal characteristics are demonstrated in Section 3.

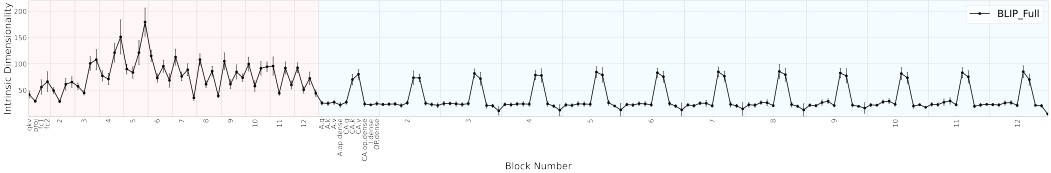

Figure 1: ID variations of the BLIP pre-training model ID across layers using the TwoNN estimator with 2,000 samples. Error bars are the standard deviation of the ID. We label the layers in the first block of each modality, and subsequent blocks follow the same repeating design. A and CA denote attention and cross-attention, respectively.

## 3 UNDERSTANDING CROSS-MODALITY LEARNING VIA IDS

### 3.1 IS THERE A HUNCHBACK IN TRANSFORMER-BASED VISUAL REPRESENTATIONS?

Ansuini et al. (2019); Muratore et al. (2022) have shown that visual representations in CNN-based models (e.g., VGG and ResNet) exhibit a typical hunchback shape in terms of ID variation across layers. However, it is unclear if this pattern holds true for Transformer-based visual models. We compare the estimated IDs of two Transformer-based models, including BLIP ViT (Li et al., 2022) and VLP (Zhou et al., 2020), to the CNN-based models, including VGG-16 (Simonyan & Zisserman, 2015) and ResNet-152 (He et al., 2016).

BLIP and VLP are implemented by dual-stream and single-stream multimodal learning paradigms. They are adopted to eliminate the impact of modality fusion methods. Since the significant structural and layer number differences among networks, they are compared on the relative depths. To make a deeper analysis, the IDs are separated by layer types. The results are shown in Figure 2.

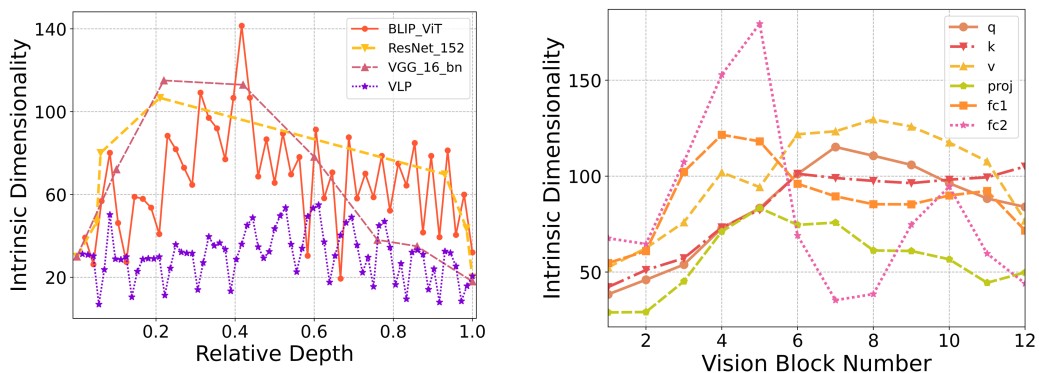

Figure 2: **1)** ID comparisons of BLIP-ViT, VLP, VGG-16-bn, and ResNet-152 in relative depth. **2)** Comparing IDs by layer types for BLIP-ViT model. q, k, and v denote query, key, and value layers in attention. proj and fc denote the projection layer and fully connected layers.

Comparing the BLIP IDs with VGG and ResNet, their variations show similar hunchback-shaped profiles. However, the peak location of BLIP lags slightly behind ResNet and VGG, moving from 0.2 to 0.4. VLP's hunchback is positioned further to the right side, and its values are much lower compared to other models. Overall, Transformer-based models have a wider range of IDs than

CNN-based models. The delayed peak and lower value of Transformer-based models' IDs can be explained by integrating textual features, particularly in the single-stream VLP model which fuses visual and language at the beginning. As described in Section 3.2, language representations have consistent and lower IDs.

We have three main observations according to Figure 2: **1)** All of each layer show hunchback profiles but with different distributions. **2)** For attention layers (q, k, and v), there is a hunchback but its peak is lagged. **3)** For MLP layers (fc1 and fc2), there are two hunchbacks, while the latter one is lower. These observations further verify the aforementioned explanations of multimodal learning. That is, the combination of vision and language representations can be described by the overlay of IDs for each modality.

## 3.2 STATISTICS OF ID FOR LANGUAGE REPRESENTATIONS

A language block consists of two attention modules: self-attention and cross-attention, as well as two feed-forward modules. Each attention module is composed of query, key, value, and fully connected layers. On the other hand, the feed-forward modules only comprise a fully connected layer.

The estimation of IDs for language representations follows the same procedure as that of visual representations. As illustrated in Figure 3, Most ID values for language layers are low, typically ranging from 5 to 30. However, the k and v layers of cross-attention exhibit ID values ranging from 70 to 90. Despite being located in the language model, the inputs to k and v layers are visual representations. Compared to visual modality, the ID values of language representations are lower and tend to remain stable across layers.

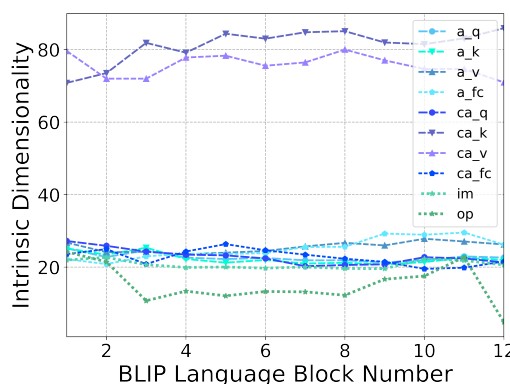

Figure 3: IDs of each language layer in BLIP. a, ca, and ff denote attention, cross attention, and feed-forward. q, k, and v denote query, key, and value. im and op denote intermediate and output.

## 3.3 INTERPRETING CROSS-MODAL ATTENTION VIA IDS

A common cross-attention mechanism for visual captioning tasks involves a three-step process: Firstly, similarity scores are computed between representations of vision and language; Secondly, attention weights are applied to these similarity scores to obtain a weighted vision representation. Finally, the vision representation is projected onto a word embedding space. Although this process provides an effective way of visual and language integration, the validity of the inner similarity estimation is unclear. Liu et al. (2021b); Hoover et al. (2019) use visualization techniques to examine cross-modality alignment, but these methods are insufficient for providing an objective and thorough evaluation. We analyze these three steps by investigating ID variations.

The first step is projecting the language representation and the visual representation to query (q) and key (k) layers. The language representations (q) lie on a low-dimensional manifold since the language ID is typically around 25. On the other hand, the IDs of the visual representations (k) do not decrease but rather increase from 40 (ID of the last visual layer) to 80. This implies that the k layer expands the visual representation to a higher-dimensional manifold that does not coincide with the language manifold. From such a perspective of ID, cross-modal similarity estimation can be viewed as a process of embedding the low-dimensional language representation into the high-dimensional visual manifold. However, this process may not be effective as the IDs of k layers show that the manifolds of these two modalities are far apart, even though the embedding dimensions of q and k are both 728.

The second step of the cross-attention mechanism involves projecting the visual representation to a value (v) layer which is the output of the cross-attention. This output representation is conceptually the closest to a cross-modal representation since it is directly supervised by the caption loss. As illustrated in Figure 3, the IDs of the v layers (the light purple line with triangles) are slightly lower than those of the k layer (the deep purple line with triangles) but still much higher than those of other layers.

The third step involves a fully connected layer that projects the cross-modal representation to a language representation. The cross-modal representation is reduced to a low-dimensional manifold, which has a similar ID with other pure language layers.

Overall, cross-attention can be viewed as a process that initially rises and then declines in intrinsic dimension. The final ID value is heavily impacted by the output modality. Typically, the visual modality yields a higher ID than language. These discoveries suggest ways to improve cross-modal attention mechanisms. For instance, by increasing the overlap between two manifolds, it is possible to achieve better alignment of vision and language representations.

# 4  PRUNING MULTIMODAL PRE-TRAINING WITH ID METRIC

As mentioned in Section 3, it is widely recognized that the ID is noticeably smaller than the embedding dimensions, often by a factor of ten or even a hundred. Despite feature dimensions ranging from 512 to 2048, their corresponding IDs typically fall below 180 in the BLIP model. In light of this, we explore the possibility of leveraging the ID metric for model pruning.

High-value IDs can be the results of two opposite conditions. One is that the training data contains so complex information that requires more dimensions to describe it. Another one is that the network learns meaningless or disordered representations, such as noise or adversarial perturbations, which lead to poor generalization. The pruning strategies for these two scenarios are completely opposite, and it depends on the features the "redundant" weights correspond to. If it is the former, then a higher ID indicates greater importance and less pruning, while if it is the latter, the opposite is true. Both views present supporting evidence (Muratore et al., 2022; Ankner et al., 2022). However, there is a lack of systematic studies currently, particularly with respect to targeting multimodal representations. To investigate the relationships among pruning, ID, and model performance, we arrange a series of experiments with different pruning strategies on BLIP pre-training.

**Implementation details.** We conduct our experiments using PyTorch on 4 NVIDIA GeForce GTX 3090 GPUs. The number of samples is 2,000, using TwoNN for ID estimation. The multimodal pre-train model used for pruning is BLIP with ViT-B/16. We perform pruning over the course of 5 epochs. During the first epoch, we train the model without pruning. In epochs 2 to 4, we use the cubic schedule (Sanh et al., 2020) to control the pruning rate at each step. The target pruning rate will be reached by the end of the fourth epoch. In the fifth epoch, the entire model is iteratively pruned at the target pruning ratio.

**Datasets**. We evaluate the performance of pruned models on image captioning task with MSCOCO datasets (Lin et al., 2014). MSCOCO is a large-scale dataset for object detection, segmentation, keypoint detection, and captioning. It covers 80 object categories and 91 stuff categories. We follow the standard splits of MSCOCO 2017, which uses 118K images for training and 5K images for both validation and testing. Each image is paired with 5 human annotated captions. We use five common metrics for evaluation: CIDEr (Vedantam et al., 2015), BLEU (Papineni et al., 2002), METEOR (Banerjee & Lavie, 2005), ROUGE (Lin, 2004), and SPICE (Anderson et al., 2016).

## 4.1  DOES PRUNING REDUCE OR INCREASE THE ID?

Muratore et al. (2022) argues that pruning luminosity and contrast information in visual representations increases the ID value, while Ankner et al. (2022) argues that the prunability of the neural network decreases as the ID increases. To resolve this discrepancy, we empirically verify it with two important metrics for weight pruning.

**Magnitude** (Zhu & Gupta, 2017) is a gradual pruning method that parameters with small magnitude are pruned. The pruning is iteratively conducted for 5 epochs. **Magnitude w/o finetune** uses the same metric but with only one-time pruning without fine-tuning.

**Sensitivity** (Zhang et al., 2022) is an iterative pruning method that takes both gradient and uncertainty into consideration for importance evaluation. It is fine-tuned for 5 epochs after pruning.

Table 1 and Figure 4.1 show the model performance across different metrics and the corresponding IDs when the pruning ratio is 80%, respectively. In most layers, the full BLIP model has larger ID values compared to other pruned models. On the other hand, the Mag w/o ft model has the smallest ID values across most layers. Accordingly, the full BLIP model exhibits the best model performance, while the Mag w/o ft model has the worst performance. When comparing the ID values of Mag and Sens models, instability is observed across layers. In the vision layers, Mag's IDs are considerably larger than Sens's in the first three blocks. However, after the hunchback peak, the IDs of Sens surpass Mag's. In the language layers, the IDs of both models are mostly close to each other, but Sens's peak ID value is larger than Mag's. Overall, we have the following observations according to the comparisons between IDs and the performance of pruned models:

**1)** Pure pruning significantly decreases IDs of almost all layers, while fine-tuning increases IDs.

**2)** The vision modality has more significant ID decreases than the language modality.

**3)** ID values have a positive correlation with model performance but not in direct proportion. We argue that the maximum ID is a more critical indicator for performance prediction, which is against the observation that the ID of the last latent layer indicates model performance (Ansuini et al., 2019).

Table 1: Model performance using different weight importance metrics when pruning ratio is 80%. BLIP is the full model before pruning, Mag, Mag w/o ft, and Sens are after pruning models. Mag, Sens, and B denote Magnitude, Sensitivity, and BLEU.

| Model | CIDEr | B@1 | B@2 | B@3 | B@4 | METEOR | ROUGE | SPICE |
|---|---|---|---|---|---|---|---|---|
| Full BLIP | 133.3 | 78.9 | 63.7 | 50.5 | 39.7 | 30.9 | 60.0 | 23.8 |
| Mag w/o ft | 0.4 | 9.0 | 0.2 | $5^{-7}$ | $9^{-10}$ | 1.7 | 9.2 | 0.0 |
| Mag | 77.6 | 65.1 | 47.7 | 34.9 | 25.8 | 22.7 | 49.3 | 15.5 |
| Sens | 124.1 | 76.9 | 61.5 | 48.2 | 37.6 | 29.6 | 58.5 | 22.6 |

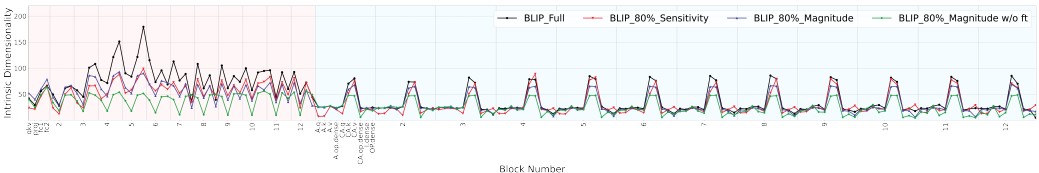

Figure 4: ID variations of pruned models with different importance metrics: Magnitude (Zhu & Gupta, 2017), Magnitude without finetuning, and Sensitivity (Zhang et al., 2022), with 80% pruning ratio on BLIP pre-trianing.

## 4.2 CAN ID PREDICT LAYER IMPORTANCE?

In Section 4.1, we study the impact of pruning on ID values. On the contrary, in this section, we investigate how incorporating ID in the importance metric affects the performance of pruned models. The PLATON (Zhang et al., 2022) proposes a sensitivity metric to measure the importance of each weight. We multiply ID, as the layer importance, with the Sens metric to verify whether it improves the pruning performance. Also, we provide results of Sens/ID to analyze the effectiveness of ID for weight pruning. Figure 5 and Table 2 show the ID value and corresponding performance comparisons, respectively.

In Table 2, it is evident that using the Sens*ID pruning strategy leads to better performance results for all evaluation metrics in comparison to using the original Sens metric. However, the Sens/ID strategy results in a significant decrease in performance. Figure 5 shows that Sens*ID considerably increases the IDs of all visual representations, including all layers in the visual model and the k and v layers

Table 2: Performance comparison with different weight importance metrics: Sens (Zhang et al., 2022), Sens*ID, and Sens/ID at 80% pruning ratio

| Model | CIDEr | B@1 | B@2 | B@3 | B@4 | METEOR | ROUGE | SPICE |
|---|---|---|---|---|---|---|---|---|
| Full BLIP | 133.3 | 78.9 | 63.7 | 50.5 | 39.7 | 30.9 | 60.0 | 23.8 |
| Sens | 124.1 | 76.9 | 61.5 | 48.2 | 37.6 | 29.6 | 58.5 | 22.6 |
| Sens*ID | 129.2 | 78.7 | 63.6 | 50.1 | 39.1 | 30.2 | 59.4 | 23.3 |
| Sens/ID | 75.0 | 65.0 | 47.2 | 34.2 | 25.2 | 22.3 | 48.9 | 15.1 |

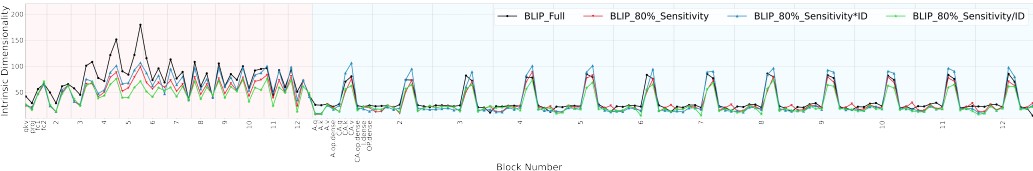

Figure 5: ID variations of pruned BLIP models with different importance metrics: Sensitivity, Sensitivity*ID, and Sensitivity/ID with 80% pruning ratio.

in the language model. On the other hand, the IDs of pure language representations decrease due to the multiplication of IDs. Based on these observations, it can be concluded that incorporating ID improves the importance evaluation of weights, thereby enhancing the overall pruning performance.

### 4.3 DO VISION AND LANGUAGE CONTRIBUTE DIFFERENT PRUNABILITY?

To better understand the contribution of each modality to the overall performance of the pruned model, we assign the same pruning ratio to different modalities. First, we examine the pruning upper bounds of different modalities. Specifically, we conducted pruning experiments on BLIP's single-modality (V and L) and the entire network (V+L) at various pruning rates (20%, 40%, 70%, 80%, 90%, and 95%). The CIDEr metric is used to evaluate the performance of the pruned model. The model performances are shown in Figure 6. It is observed that a pruning ratio of over 70% resulted in a significant decline in the efficacy of most models, except for the language model pruning alone with the Sens*ID metric (the light blue line with triangles). The models that only prune the visual modality exhibit the fastest decay, whereas the models that only prune the language modalities experience only a 3-6 drop in CIDEr compared to the full BLIP model, even when the pruning ratio is as high as 95%. Overall, the language models have a much higher upper-bound pruning ratio with different importance metrics.

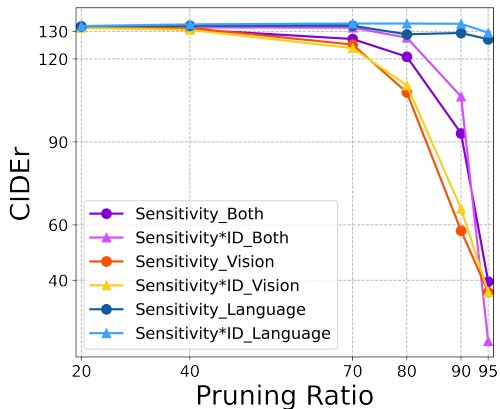

Figure 6: Performance comparison when pruning ratio ranges from 20% to 95% for individually pruning vision, language, and the entire model, respectively.

Figure 7 shows the change of IDs when pruning only one modality. Pruning any one modality alone will cause the ID of the other modality to change. Intriguingly, only pruning vision layers leads to ID decrease for both vision and language layers, while only pruning language layers leads to ID increase in vision layers (also including k and v layers in cross-attention) but a decrease in pure language layers.

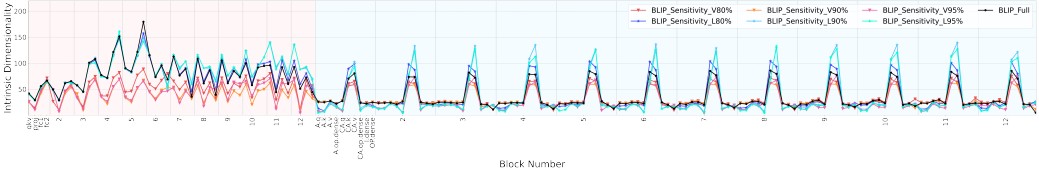

Figure 7: When one modality is pruned (red for vision, blue for language) at 80%, 90%, and 95% ratios, its effect on the ID changes of both modalities.

Table 3 presents a detailed comparison of the performances between pruning single modality and the entire model. In general, it is observed a decline in model performance after pruning. However, pruning only the language model with an 80% pruning ratio results in an improvement in overall model performance across several evaluation metrics. Comparing Sens and Sens*ID, the improvement is significant for pruning vision, language, and both modalities. Comparing Sens and Sens/ID, the model performance of Sens/ID degrades significantly for only pruning vision models, whereas the performances of pruning language models are similar with both Sens and Sens/ID metrics. Based on the observations, we argue that language representations are more robust for pre-training. Even if some important weights in language layers are pruned, performance can be restored by fine-tuning, which leads to a higher lower bound of performance. On the contrary, visual representation is more sensitive and fragile, and incorrect importance metrics can noticeably degrade model performance.

Table 3: When pruning vision model (V), language model (L), and the entire model (V+L) individually at 80% pruning ratio, performance comparison of pruned models over multiple metrics.

| Model | Prune | CIDEr | B@1 | B@2 | B@3 | B@4 | METEOR | ROUGE | SPICE |
|---|---|---|---|---|---|---|---|---|---|
| Full BLIP | None | **133.3** | 78.9 | 63.7 | 50.5 | **39.7** | **30.9** | **60.0** | **23.8** |
| Sens | V | 107.9 | 72.8 | 56.5 | 43.4 | 33.4 | 27.5 | 55.3 | 20.4 |
| | L | 128.9 | 78.6 | 63.6 | 50.1 | 39.0 | 30.0 | 59.3 | 23.0 |
| | V+L | 124.1 | 76.9 | 61.5 | 48.2 | 37.6 | 29.6 | 58.5 | 22.6 |
| Sens*ID | V | 110.5 | 73.2 | 57.1 | 44.0 | 33.8 | 27.9 | 55.8 | 20.7 |
| | L | 132.8 | **79.4** | **64.3** | **50.9** | **39.7** | 30.6 | 59.9 | **23.8** |
| | V+L | 129.2 | 78.7 | 63.6 | 50.1 | 39.1 | 30.2 | 59.4 | 23.3 |
| Sens/ID | V | 101.1 | 71.0 | 54.3 | 41.3 | 31.5 | 26.0 | 54.0 | 19.4 |
| | L | 128.7 | 78.7 | 63.7 | 50.0 | 38.7 | 28.9 | 59.3 | 22.9 |
| | V+L | 75.0 | 65.0 | 47.2 | 34.2 | 25.2 | 22.3 | 48.9 | 15.1 |

## 5 CONCLUSION

This work delves into the ID characteristics of the BLIP large-scale pretraining model for multi-modal learning. Also, the study investigates the potential of utilizing ID variations to determine the significance of each layer. Experimental results indicate that the ID of visual modalities generally exhibits a hunchback profile and has a broad range of ID values (29-180), while language modalities have a consistent distribution of ID values, with a range of 5-30 on each layer. The two modalities' representations are semantically integrated through multi-modal learning, which can be represented by combining their ID distributions. ID can be used to evaluate the importance level of network layers and enhance pruning performance. Moreover, our work finds that language modality in large-scale BLIP pre-training has more redundant weights but is robust to pruning, while vision modality is sensitive but has a greater impact on overall model performance.

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
