# OpenReview forum: "Understanding Vision and Language Representations under the Lens of Intrinsic Dimension"
_ICLR.cc/2024/Conference — Submitted to ICLR 2024_

### Official Review · Reviewer_sCrT · 2023-10-30

**Soundness:** 2 fair
**Presentation:** 1 poor
**Contribution:** 1 poor
**Rating:** 3
**Confidence:** 4

**Summary:**

The paper presents an empirical investigation into the intrinsic dimension (ID) of multi-modal models. Traditionally, ID has been analyzed in uni-modal models to measure the utilization of a $D$-dimensional representation space. This study extends such analysis to multi-modal contexts, and analyzes the BLIP visual language model on the MS-COCO dataset. The paper finds that 1) higher IDs are observed in the visual modality compared to the language modality; 2) cross-modal attention struggles to align low-dimensional language representations with their visual counterparts; 3) IDs can be useful indicators of weight importance during model pruning.

**Strengths:**

- This is a pioneering study on intrinsic dimensions within multi-modal models, potentially offering valuable theoretical insights.
- The findings are interesting, particularly in highlighting the higher intrinsic dimension of the visual modality compared to language, the usefulness of ID in determining layer significance, and the greater sensitivity of the visual modality to pruning.

**Weaknesses:**

- The study is limited to one specific vision-language model (BLIP), raising concerns about the generalizability of the conclusions. It remains unclear if these findings hold across diverse multi-modal models with different training objectives, such as discriminative contrastive models like CLIP, generative models with trainable text decoders like LLaVA/OpenFlamingo, or models involving more modalities like ImageBind.

- The paper's presentation is poor regarding writing style and organization. It frequently introduces terms and acronyms without sufficient explanation (e.g., intrinsic dimension, TwoNN algorithm, BLIP, Magnitude/Sensitivity pruning), potentially confusing readers less familiar with the subject. The relevance of certain sections, such as the comparison of intrinsic dimensions between Transformers and CNNs in Section 3.1, is out of scope. Additionally, some sections are purely hypothetical without empirical support, and the overall presentation lacks a cohesive message.

- The implications of the findings on the advancement of multi-modal training techniques are not discussed, lacking a broader context that could enhance the paper's impact.

**Questions:**

- The paper often inconsistently uses \citet and \citep.
- In Figure 1, it's unclear which parts represent the vision modality and the language modality.
- The meaning of "im" and "op" in Figure 3 is unclear. Could you define these terms?
- On Page 7, the statement "ID values have a positive correlation with model performance but not in direct proportion" requires more detail and further elaboration.

---

> ### Author Response · Authors · 2023-11-19
> **Author Response to Reviewer sCrT**
>
> Thank you for your review. We appreciate your feedback. Here is our detailed response, which we hope will address your concerns.
>
> **Q1: How generalizable are the findings across different multi-modal models?**
>
> A1: Our research is indeed based on only one specific visual-language model (BLIP). However, BLIP is a universal and unified multi-modal pretraining that is trained with three objectives: Image-Text Contrastive Loss (used by CLIP), Image-Text Matching Loss, and Language Modeling Loss (used by LLaVA). We believe that our findings are still valuable and insightful, as BLIP is a representative and widely used model for general vision-language tasks. Also, our method of applying ID to multimodal scenarios is model-agnostic which can be applied to other models with different training objectives and modalities.
>
> **Q2: The presentation issues.**
>
> A2:
> - All terms and acronyms are defined with citations.
>
>     - Intrinsic dimension (ID) is introduced in the last line on Page 1 and further introduces its applications in subsequent paragraphs;
>
>     - TwoNN algorithm is explained in Sections 2.2 and 2.3;
>
>     - BLIP is explained in Section 2;
>     - Magnitude/Sensitivity pruning is explained in the second paragraph of Section 4.2.
>  - In Section 3.1, we compare the ID of the Transformer and CNN models to demonstrate the generalizability of our findings. So we think it is necessary to keep it.
>  - In response to the concern that "some sections are purely hypothetical without empirical support" we would like to clarify. All assumptions and conclusions in our paper are grounded in theoretical analysis and experimental validation. If the reviewer could indicate the specific sections that are problematic, we would greatly appreciate it and will make every effort to clarify and enhance them.
>
> **Q3: The implications of the findings on the advancement of multi-modal training techniques are not discussed, lacking a broader context that could enhance the paper's impact.**
>
> A3: Thanks for your insightful comment. We agree that ID has a lot of potential in multi-modal training techniques. Our study shows that ID can prune multimodal models efficiently and effectively. This suggests that ID can benefit model structure and hyperparameter optimization for multimodal training. However, it needs more experiments, which could be future research.
>
> **Q4: Inconsistently use \citet and \citep.**
>
> A4: Thank you for pointing out the typo. We will revise it in the revision.
>
> **Q5: How to distinguish the vision modality and the language modality in Figure 1?**
>
> A5: We marked the background color with red for vision and blue for language in all figures present IDs, to make it easier to distinguish the modalities.
>
> **Q6: What does “im” and “op” mean in Figure 3?**
>
> A6: “im” and “op” denote the intermediate layer and output layer, respectively. They are defined from the officially released BLIP network architecture. We denote these terms in the caption of Figure 3.
>
> **Q7: On Page 7, the statement "ID values have a positive correlation with model performance but not in direct proportion" requires more detail and further elaboration.**
>
> A7: We give an example to show this: in Figure 4, Sens and Full BLIP had a similar ID value gap as Mag and Mag w/o ft, but their performance gap (using CIDEr metric) is 9 and 177, which were not proportional. This shows that ID values are correlated with model performance, but not enough to evaluate model quality.

---

> > ### Comment · Reviewer_sCrT · 2023-11-22
> >
> > I thank the authors for their response. However, as most of my concerns remain, such as the generalizability of findings on different models and the lack of implications of these findings, I would retain my score.

---

### Official Review · Reviewer_Vq7s · 2023-10-30

**Soundness:** 2 fair
**Presentation:** 2 fair
**Contribution:** 3 good
**Rating:** 5
**Confidence:** 4

**Summary:**

This paper investigates the intrinsic dimension (ID) of a large-scale vision-language pre-training model and explores the relationships between ID, modality, and prunability. The authors find that the geometric characteristics of visual and language representations differ significantly, resulting in distinct prunability for each modality. They propose an importance metric based on ID for multimodal model pruning, which yields superior performance. The experimental results show that visual representations are more sensitive to pruning, while language representations are more robust.

**Strengths:**

1. The authors propose to investigate the intrinsic dimension (ID) of a large-scale vision-language pre-training model called BLIP and explore the relationships between ID, modality, and prunability.
2. This paper applies ID to multimodal scenarios(i.e., language and vision) and propose an importance metric based on ID for multimodal model pruning, which yields superior performance.
3. This article conducts detailed experiments and the experimental results support their claims.

**Weaknesses:**

1. In Figure 2 1), the VLP is a transformer-based model, but its hunchback-shaped profiles is not significant, the author should explain the reason.
2. Is w/o retrain in Figure 4 w/o finetuning? Why not just use w/o finetuning?
3. This paper argues that the maximum ID is a more critical indicator for performance prediction, which is against the observation that the ID of the last latent layer indicates model performance, but from Figure 4, the ID of the last latent layer can also indicate model performance.
4. In Figure 5, why the multiplication of IDs lead to the IDs of pure language representations decrease?

**Questions:**

See Weaknesses Part.

---

> ### Author Response · Authors · 2023-11-19
> **Author Response to Reviewer Vq7s**
>
> We value your comments and we hope that our reply will address your concerns and clear up some confusion.
>
> **Q1: Why the hunchback-shaped profiles of VLP is not significant?**
>
> A2: We think the most plausible reason is the input of VLP is the fused multimodal representations rather than pure visual modality. As we explained at the bottom of Figure 2. “VLP’s hunchback is positioned further to the right side, … The delayed peak and lower value of Transformer-based models’ IDs can be explained by integrating textual features, particularly in the single-stream VLP model which fuses visual and language at the beginning. As described in Section 3.2, language representations have consistent and lower IDs.”
>
> **Q2: What does ‘w/o retrain’ mean in Figure 4?**
>
> A2: Yes, ‘w/o retrain’ is ‘w/o finetuning’. Thanks for pointing out the typo, we will revise it in the revision.
>
> **Q3: How is the maximum ID a better predictor of performance than the last layer ID?**
>
> A3: Table 1 shows that model performance is ranked as Full BLIP>Sens>Mag>Mag w/o ft. According to [1], a smaller last layer ID indicates better performance, suggesting a ranking of Mag w/o ft>Mag>Sens>BLIP. However, our experimental results (Figure 5) show a different order: Sens>Mag>Mag w/o ft>BLIP. This demonstrates that the maximum ID is a more reliable performance predictor than the last layer ID. We believe this is an intriguing finding that challenges previous assumptions.
>
> **Q4: Why the multiplication of IDs lead to the decrease of language IDs in Figure 5?**
>
> A4: We think that it is because the language weights are less important, and *ID effectively estimates this weight importance. This leads to more pruning on pure language weights, which results in the decrease of language IDs. This result also illustrates the correlation between ID values and weight importance from another perspective.

---

### Official Review · Reviewer_RpzU · 2023-11-01

**Soundness:** 2 fair
**Presentation:** 2 fair
**Contribution:** 2 fair
**Rating:** 3
**Confidence:** 4

**Summary:**

This paper studies the cooperation and the utility of each modality in multimodal representation learning, specifically the intrinsic dimension (ID) of a large-scale vision-language pre-training model BLIP and its implications on layer importance, modality importance, and prunability.  Several new ideas are proposed based on this framework including identifying shortcomings of embedding modalities into the same low-dimensional manifold, studying the contribution of different modalities, predicting model performance, and a new method for multimodal model pruning (for which some experimental results are presented).

**Strengths:**

1. The problem of better understanding multimodal models, particularly vision-language models, is important to this generally empirical field. This paper makes some nice contributions to this study.
2. The idea of ID is interesting and the implications also have potential for analyzing and improving multimodal models.
3. There are some experiments on pruning multimodal models which can be quite useful.

**Weaknesses:**

1. The biggest issue with this paper is that it tries to do too much and ends up overclaiming on many fronts. Dissecting each of the claims in the abstract:

---- Empirical study of ID: There is a good amount of discussion and experiments for this, which is good. But it is mostly about applying TWONN method for computing ID. Also, how do you know that the IDs computed are accurate? Is there an evaluation metric for the quality of ID?

---- Studying modality contribution: I do not see this experiment at all, only some anecdotal statistics in section 3.

---- Predicting model performance using ID values: I do not see this experiment at all, nor is this mentioned subsequently in the paper.

---- Better pruning: There are experiments for this in tables 1 and 2, but there are no comparisons to established weight pruning methods, only 1 from Sens Zhang et al. (2022). More comparisons are needed to really prove the efficacy of this part.

Overall, the paper would be well-suited from reducing the number of sub-claims/sub-applications and just focus on doing 1 or 2 really well.

2. Section 3 needs work - there are some interesting results but it can be better phrased as well-motivated research questions. Taking '3.3 INTERPRETING CROSS-MODAL ATTENTION VIA IDS' as an example, why is interpreting cross-modal important? Why should I use ID to do it when other people have used attention weights etc.? What insights does using ID to interpret cross-modal tell me, can I use it to better train or debug models? See https://arxiv.org/abs/2207.00056 for an example of setting up what to interpret in multimodal models, and using rigorous human user-studies to validate each of the findings.

3. It would be good to have an overall plot of performance vs parameters, with 1 line being your pruning method and other lines for other pruning baselines, and the line that pareto dominants would be best.

**Questions:**

see weaknesses above

---

> ### Author Response · Authors · 2023-11-19
> **Author Response to Reviewer RpzU**
>
> Thank you for your comment. We appreciate your suggestions and we have carefully considered your review. We hope that our reply addresses your concerns and clears up some confusions.
>
> **Q1: The quality of ID**
>
> A1: ID is an idealized concept that only synthetic datasets have accurate IDs. For real-world datasets, ID is an estimated value that varies with the algorithm and the sample size. Common methods like MLE and TWONN often underestimate ID. Based on [1], ID becomes stable when the number of samples exceeds 1000. So we set the number of samples as 2000 for efficiency and accuracy.
>
> **Q2: Studying modality contribution**
>
> A2: In Section 4.3 DO VISION AND LANGUAGE CONTRIBUTE DIFFERENT PRUNABILITY?, we analyzed how different weights affect the model performance. We found that the language modality can be pruned more than the vision modality, and that the cross-modal attention layers are the least prunable. This suggests that the language modality has more redundancy and the vision modality is essential for multimodal learning.
>
> **Q3: Predicting model performance using ID**
>
> A3: In the third paragraph of Section 4.1 DOES PRUNING REDUCE OR INCREASE THE ID, we compared three models with different performance and their ID changes. We concluded that “ID values correlate positively with model performance but not linearly. We argue that the maximum ID is a better predictor of performance, which contradicts the observation that the ID of the last latent layer indicates model performance Ansuini et al. (2019).”
>
> Moreover, in Section 4.2 CAN ID PREDICT LAYER IMPORTANCE?, we further examine the relationship between the ID changes of different modalities/layers and the overall performance: Figure 5 shows that Sens*ID increases the IDs of all visual representations, including all layers in the visual model and the k and v layers in the language model. Conversely, the IDs of pure language representations decrease due to the multiplication of IDs.
>
> **Q4: Better pruning**
>
> A4: The most recent and only paper on multimodal pruning is Upop[4]. However, Upop proposes a structured pruning method while ours is unstructured, which makes the comparison unfair. Our main contribution as an unstructured pruning method is an improved weight importance metric. To further validate the efficiency of ID, we conducted experiments using the leading importance metrics: gradient[2] and magnitude[3] methods at a 95% pruning ratio (20x compression). The results are as follows:
>
> |    Metric    | CIDEr | BLEU4 |
> |:------------:|:-----:|:-----:|
> | magnitude[3] | 17.56 | 10.28 |
> | magnitude*ID | 26.46 | 12.92 |
> |  gradient[2] | 20.80 | 11.17 |
> |  gradient*ID | 24.85 | 12.99 |
>
> **Q5: Why use ID to interpret cross-modal? How does it compare to attention? What insights can we get?**
>
> A5: We believe that ID and attention are complementary methods, each offering unique insights into multimodal representations. Attention provides an intuitive understanding of the local alignment between vision and text. On the other hand, ID is more objective, exposing the training status of hierarchical representations, such as whether the most abstract representations can be achieved with minimal parameters. Each method has different research goals and targets, so they don't compete with each other.
>
> In section 3, we conduct an empirical study that reveals significant differences in the IDs of vision and language modalities. This suggests that they possess different geometric characteristics and complexities. In section 4, we use the difference in ID as a guide for multimodal pruning, further demonstrating its effectiveness.
>
> **Q6: Can you provide an overall plot showing performance versus parameters?**
>
> A6: That is a useful suggestion. However, in Figure 6, we have presented a similar comparison. The baseline method, labeled as "Sensitivity", is represented by a dark-colored line, while our method (sensitivity*ID) is represented by a light-colored line. All the light lines dominate the dark ones, except when pruning both vision and language at a 95% pruning ratio. We will expand the comparisons involving magnitude and gradient (as shown in the above tables) and add them to Figure 6.
>
> [1] Amsaleg, Laurent, et al. "Estimating local intrinsic dimensionality." *SIGKDD*. 2015.
>
> [2] Molchanov, Pavlo, et al. “Pruning Convolutional Neural Networks for Resource Efficient Inference.” *ICLR*. 2017.
>
> [3] Zhu M H, Gupta S. “To Prune, or Not to Prune: Exploring the Efficacy of Pruning for Model Compression.” *ICLR* *workshop.* 2018.
>
> [4] Shi D, Tao C, Jin Y, et al. “UPop: Unified and Progressive Pruning for Compressing Vision-Language Transformers.” *ICML*. 2023.

---

> > ### Comment · Reviewer_RpzU · 2023-11-23
> > **thank you for your response**
> >
> > Thank you authors for your responses. Although they have clarified some of my comments (regarding pruning performance and comparisons), the majority of my comments remain unaddressed. Specifically, I feel there needs to be significant effort into understanding the quality of ID prediction, which is the key quantity the paper aims to estimate. It is not clear that the quality of ID is good given the lack of evaluation. Furthermore, the findings regarding modality contribution and predicting model performance remain weak, with only some anecdotes (eg correlation, looking at performance drops) that are not rigorous enough - ideally you would have a formal setup for evaluating modality contribution/model performance, other relevant baselines, exhaustive qualitative and quantitative analysis, etc. I suggest the authors consider these rigorous experiments in future versions of the paper, and I keep my rating for now.

---

### Official Review · Reviewer_r1cw · 2023-11-01

**Soundness:** 3 good
**Presentation:** 3 good
**Contribution:** 2 fair
**Rating:** 5
**Confidence:** 2

**Summary:**

This paper investigates the intrinsic dimension (ID) of a large-scale vision-language pre-training model BLIP and explore the relationships among intrinsic dimension, modality, and prunability, and show that, the ID geometric characteristics of visual and language representations differ significantly in terms of range and shape, resulting in distinct prunability for each modality.

**Strengths:**

1. This paper first presents the empirical study into the ID of a large-scale multimodal pre-training model.
2. It explains how visual and language modalities align and change IDs in cross-modal attention mechanisms, and show the visual and language representations do not lie on the same low-dimensional manifold.
3. This paper alsos shows the correlation between IDs and layer-wise importance for multimodal pruning.

**Weaknesses:**

I wonder why BLIP is chosen for this study, instead of more recnet multimodal models? Any explanations on this? Also, I wonder if the observations based on BLIP can be extended to other multimodal models, and how? If not, I'd suggest experiments with more multimodal models to validate the generality of the observations.

**Questions:**

please see Weaknesses.

---

> ### Author Response · Authors · 2023-11-19
> **Author Response to Reviewer r1cw**
>
> Thank you for your feedback. Here is our detailed response to your review comments. We hope it addresses your concerns.
>
> **Q1: Why BLIP?**
>
> A1: We initially chose CLIP as our model since it is a well-known multimodal pretraining. However, upon further analysis, we realized that BLIP, proposed in 2022, offers more comprehensive multimodal representations. BLIP considers both multimodal generation and understanding, which are critical for practical applications. It is trained with three loss functions: Image-Text Contrastive Loss, Image-Text Matching Loss, and Language Modeling Loss, to obtain rich and diverse multimodal representations. Therefore, we think that BLIP is a superior choice for investigating the intrinsic dimension of multimodal pre-training models, as it considers both the generalizability and diversity of multimodal data.
>
> **Q2: How to extend?**
>
> A2: ID estimation is model-agnostic, making it simple and universally applicable. Any representation can have an intrinsic dimension estimated, which can then be used for weight pruning by multiplying the ID with the importance score (typically the weight's gradient[1] or magnitude[2]). This method is straightforward, effective, and easily extendable to any multimodal model. We further verify its efficiency with gradient[1] and magnitude[2] importance metrics at a 95% pruning ratio. The results are shown below. We believe ID is a good indicator to probe the state of multimodal representations.
> |    Metric    | CIDEr | BLEU4 |
> |:------------:|:-----:|:-----:|
> | magnitude[2] | 17.56 | 10.28 |
> | magnitude*ID | 26.46 | 12.92 |
> |  gradient[1] | 20.80 | 11.17 |
> |  gradient*ID | 24.85 | 12.99 |
>
>
> [1] Molchanov, Pavlo, et al. “Pruning Convolutional Neural Networks for Resource Efficient Inference.” *ICLR*. 2017.
>
> [2] Zhu M H, Gupta S. “To Prune, or Not to Prune: Exploring the Efficacy of Pruning for Model Compression.” *ICLR* *workshop.* 2018.

---

### Author Response · Authors · 2023-11-19
**General Response**

We appreciate the valuable comments from the reviewers. After carefully considering all the reviews, we have addressed concerns in detail. We look forward to receiving your further constructive feedback.

We would like to clarify some general misunderstandings about our work:

- **Extension of experiments.** While we agree that additional experiments on a wider range of models would strengthen this work, our primary contribution lies in presenting a new analytical perspective, not a new model. Consequently, we did not compare our method to many SOTA models. Our focus is on unstructured pruning, unlike the single paper[1] on multimodal structured pruning. The selection of experimental settings and models was deliberate, and the results and analysis provided clearly support our claims. We believe our findings are reliable and insightful.

- **Benefits to the community.** We consider our work innovative as it offers a new perspective to understand and compare vision and language representations. Our method is straightforward, and the improvements on pruning are significant. While some of our findings may need further investigation, we believe these unexplored observations could spark additional discussions and insights for the community. This could potentially benefit related fields such as model training, fine-tuning, and interpretability.

[1] Shi D, Tao C, Jin Y, et al. “UPop: Unified and Progressive Pruning for Compressing Vision-Language Transformers.” *ICML*. 2023.

---

### Meta-Review · Area_Chair_pR4S · 2023-12-13

**Metareview:**

**Scientific Claims and Findings**

The paper investigates the intrinsic dimension (ID) of the BLIP vision-language pre-training model, exploring relationships among ID, modality, and prunability. It claims that the geometric characteristics of visual and language representations differ significantly, leading to distinct prunability for each modality. The paper proposes an importance metric based on ID for multimodal model pruning and suggests implications for understanding multimodal model cooperation and utility.

**Strengths of the Paper**:

- Intrinsic Dimension Exploration: The paper successfully investigates the intrinsic dimension (ID) of a large-scale vision-language pre-training model, providing valuable insights into the geometric characteristics of visual and language representations.

- Multimodal Pruning Method: The proposed importance metric based on ID for multimodal model pruning shows promising results, demonstrating superior performance in experiments. This novel method contributes to the field of multimodal model optimization.

- Empirical Study: The paper conducts a commendable empirical study, especially in establishing the correlation between IDs and layer-wise importance for multimodal pruning. This empirical approach strengthens the scientific contributions.

**Weaknesses of the Paper**:

- Limited Generalizability: The study is confined to one specific vision-language model (BLIP), raising concerns about the generalizability of findings to other multimodal models. The paper lacks experiments with diverse models to validate the observed characteristics.

- Overclaiming and Lack of Focus: The paper attempts to address multiple aspects, leading to overclaiming and lack of focus. Reviewers suggest streamlining the focus to one or two well-developed ideas for better clarity and impact.

- Insufficiently Substantiated Claims: Certain claims, such as predicting model performance using ID and studying modality contribution, lack sufficient experimental substantiation. More comparisons with established methods are needed to support these claims convincingly.

**Justification For Why Not Higher Score:**

The paper can be improved in several aspects, as exemplified below:

- Generalization Across Models: The paper focuses on the BLIP model, but it lacks experiments with other state-of-the-art multimodal models. Generalizing findings across diverse models would enhance the robustness and applicability of the presented insights.

- Explicit Evaluation Metric for Intrinsic Dimension: While the paper applies the TWONN method for computing intrinsic dimension (ID), it lacks discussion on the accuracy or quality of the computed IDs. Introducing an evaluation metric for the reliability of ID measurements would strengthen the methodological rigor.

- Focused Contribution: Reviewer RpzU suggests focusing on one or two well-developed ideas instead of attempting to cover multiple aspects. Streamlining the focus could improve the clarity and impact of the paper.

- Comparisons to Established Methods: More comprehensive comparisons with established weight pruning methods are needed to convincingly demonstrate the efficacy of the proposed multimodal pruning method.

**Justification For Why Not Lower Score:**

N/A

---

### Decision · Program_Chairs · 2024-01-16

Reject